# The emergence of modern zoogeographic regions in Asia examined through climate–dental trait association patterns

Liping Liu [1,2] ✉, Esther Galbrun [3] ✉, Hui Tang [1,4,5], Anu Kaakinen [1], Zhongshi Zhang [6], Zijian Zhang[7] & Indrė Žliobaitė [1,8]

The complex and contrasted distribution of terrestrial biota in Asia has been linked to active tectonics and dramatic climatic changes during the Neogene. However, the timings of the emergence of these distributional patterns and the underlying climatic and tectonic mechanisms remain disputed. Here, we apply a computational data analysis technique, called redescription mining, to track these spatiotemporal phenomena by studying the associations between the prevailing herbivore dental traits of mammalian communities and climatic conditions during the Neogene. Our results indicate that the modern latitudinal zoogeographic division emerged after the Middle Miocene climatic transition, and that the modern monsoonal zoogeographic pattern emerged during the late Late Miocene. Furthermore, the presence of a montane forest biodiversity hotspot in the Hengduan Mountains alongside Alpine fauna on the Tibetan Plateau suggests that the modern distribution patterns may have already existed since the Pliocene.

The Asian continent is characterized by a complex and contrasted distribution of climate and biota. Not only is there a stark north–south zoogeographic division between the Indomalayan and Palearctic realms[1,2], but also an east–west zoogeographic distributional pattern controlled by the East Asia Monsoon[3–5].

Moreover, due to the presence of the Tibetan Plateau, the Asian continent hosts montane biodiversity hotspots at low latitudes[6], of which there is no equivalent elsewhere on the planet. The diverse biotas make Asia one of the most interesting areas for studying how modern biogeographical regions emerged following tectonic and climatic changes. The emergence of the modern biogeographical regions has been linked to the topographic rise of the Tibetan Plateau[7], monsoon circulation[8–11], and transformations of the global climate[12]. The timing of these changes and their controlling mechanisms are still debated and have been key questions in the field of paleontology. The difficulty is not only the sparsity and incompleteness of fossil records, in terms of both space and time, but also the lack of methods allowing to delineate geographical regions in the past, where the distribution of fossil communities has no comparable extant relatives. Alternatively, there have been several studies attempting to track modern biotas back through time along phylogenies to shed light on their evolution, but they either provide only indirect phylogenetic evidence[13–15] or are incompatible with fossil evidence[5].

In this work, we analyze the dental traits of large herbivore communities, in order to track changes in mammal communities and their climatic contexts in the deep past. Previous works showed that the distribution of dental traits across mammalian communities correlates well with climatic conditions, in the present[16] and in the past[8,10], and

[1]Department of Geosciences and Geography, University of Helsinki, P.O. Box 64, Helsinki FI-00014, Finland. [2]Department of Palaeobiology, The Swedish Museum of Natural History, P.O. Box 50007, Stockholm SE-104 05, Sweden. [3]School of Computing, University of Eastern Finland, Technopolis, Microkatu 1, Kuopio FI-70210, Finland. [4]Climate System Research Unit, Finnish Meteorological Institute, P.O. Box 503, Helsinki FI-00101, Finland. [5]Department of Geosciences, University of Oslo, P.O. Box 1022, Oslo NO-0315, Norway. [6]Department of Atmospheric Science, School of Environmental Studies, China University of Geosciences, 388 Lumo Road, 430074 Wuhan, China. [7]Key Laboratory of Cenozoic Geology and Environment, Institute of Geology and Geophysics, Chinese Academy of Sciences, 19, Beitucheng Western Road, Chaoyang District, 100029 Beijing, China. [8]Department of Computer Science, University of Helsinki, P.O. Box 68, University of Helsinki FI-00014, Finland. ✉e-mail: liping.liu@helsinki.fi; esther.galbrun@uef.fi

can be used to build global and regional predictive models for precipitation and temperature[17,18], as well as for productivity[19] or vegetation cover[20], based on such traits. A computational data analysis methodology called redescription mining was recently tailored to biogeographic studies[21,22], aiming to identify local patterns of association between the dental traits of mammalian communities, on the one hand, and the climatic conditions, on the other hand. Here, we employ this methodology to generate and explore hypotheses about the emergence of modern zoogeographic regions. First, we extract redescriptions, i.e., local patterns of association, from high-resolution data about mammalian communities and climatic conditions across Asia in the present day. Then, we evaluate the redescriptions on data of fossil mammalian communities and modeled paleoclimatic conditions during five intervals of the Neogene, that is, since 22 million years ago. Through this lens, we examine the build-up of modern zoogeographic regions in Asia.

## Results

Redescriptions are pairs of logical statements about the values of the data variables. In our context, they capture the interplay between dental traits of mammalian communities and climatic conditions. More specifically, we consider seven dental traits and nineteen bioclimatic variables, listed in Table 1. These dental traits, introduced and studied in previous work[17,18,23], are understood to have good correlations with the encountered environmental conditions, including the climate.

A logical statement, also known as a query, specifies a range of values the involved variables might take, typically by means of

**Table 1 | List of the dental traits and bioclimatic variables**

| Dental traits variables | | |
|---|---|---|
| HYP | Average ordinated hypsodonty | |
| AL | Fraction of taxa with acute lophs | |
| OL | Fraction of taxa with obtuse lophs | |
| SF | Fraction of taxa with structural fortification of cusps | |
| OT | Fraction of taxa with flat occlusal topography | |
| OO | Fraction of taxa with exclusively obtuse lophs | |
| BU | Fraction of taxa without any lophs (bunodonts) | |
| **Bioclimatic variables** | | |
| TMeanY | bio1 | Mean annual temperature |
| TMeanRngD | bio2 | Mean diurnal range |
| TIso | bio3 | Isothermality |
| TSeason | bio4 | Temperature seasonality |
| TMaxWarmM | bio5 | Max temperature of warmest month |
| TMinColdM | bio6 | Min temperature of coldest month |
| TRngY | bio7 | Annual temperature range |
| TMeanWetQ | bio8 | Mean temperature of wettest quarter |
| TMeanDryQ | bio9 | Mean temperature of driest quarter |
| TMeanWarmQ | bio10 | Mean temperature of warmest quarter |
| TMeanColdQ | bio11 | Mean temperature of coldest quarter |
| PTotY | bio12 | Annual precipitation |
| PWetM | bio13 | Precipitation of wettest month |
| PDryM | bio14 | Precipitation of driest month |
| PSeason | bio15 | Precipitation seasonality |
| PWetQ | bio16 | Precipitation of wettest quarter |
| PDryQ | bio17 | Precipitation of driest quarter |
| PWarmQ | bio18 | Precipitation of warmest quarter |
| PColdQ | bio19 | Precipitation of coldest quarter |

Temperature and precipitation are respectively in degrees Celsius (°C) and in millimeters (mm).

thresholds. Given data recording the values of these variables at the localities within our study area, each query implicitly selects a subset of localities, those localities where the values satisfy the requirements stated in the query. For example, a query over the bioclimatic variables might require the mean annual temperature (TMeanY) to be lower than 15.4 °C, selecting the localities with colder climates. On the other hand, a query over the dental traits might require the fraction of species with structural fortification of cusps (SF) to be lower than 22.2% and the fraction of bunodont species (BU) to be lower than 35.7%, selecting the localities where the prevalence of structural fortification of cusps and the prevalence of bunodonty are both low.

A redescription then consists of a pair of queries, here respectively over dental traits and bioclimatic variables, that select similar subsets of localities, thereby capturing a pattern of association between the involved variables and value ranges. The similarity of the two subsets of localities is measured using the Jaccard coefficient, denoted as J, and generally referred to as the accuracy of the redescription, while their intersection is referred to as the support of the redescription, whose size as a percentage of the total number of localities studied is denoted as supp%. Note that redescriptions are not predictive models, the accuracy is a similarity measure calculated on the considered dataset and is not related to a prediction task. Intuitively, the closer the accuracy of a redescription is to one, the stronger the association between the corresponding conditions in the dataset. Redescription mining algorithms build and evaluate many pairs of queries, looking for those that have the highest Jaccard coefficients, while fulfilling user-defined constraints (e.g., on the complexity of the queries and the number of satisfying localities). As a major difference, redescription mining is a descriptive approach capturing patterns of associations between subsets of variables that hold locally, whereas machine-learning approaches such as regression produce a global predictive model, assuming that the same relation between the variables holds throughout the dataset.

### Characterizing modern zoogeographic regions

Having applied a redescription mining algorithm on our present-day dataset, we manually select among the obtained results nine redescriptions that have a high accuracy on that dataset and together provide a good coverage of the study area as well as of the different dental traits variables.

The selected redescriptions, denoted as rA–rI, are shown in Fig. 1. For each redescription, we list the query over dental traits variables ($q_D$), the query over bioclimatic variables ($q_C$), the accuracy (J) as well as the size of its support as a percentage of the total number of present-day localities (supp%). The status of each redescription across the study area is visualized as a map, with a dot for each present-day locality, whose color indicates whether the queries of the considered redescription are satisfied at that locality. We use the same color code throughout to represent the four possible cases: purple where both the dental traits query and the climate query are satisfied; red where the dental traits query is satisfied but the climate query is not; blue where the climate query is satisfied but the dental traits query is not; gray where neither the dental traits query nor the climate query is satisfied. The support of each redescription, defined as the set of localities where both queries are satisfied, is hence drawn in purple. Note that the queries used as example above are from an actual obtained redescription, namely rA.

We see that the redescriptions delineate areas corresponding to prominent ecoregions and notable mammalian distribution patterns in present-day Asia[2].

Among the nine redescriptions, five (rA, rB, rD, rF, and rI) outline a north–south zoogeographic pattern across the study area. Redescriptions rA, rD, and rF select regions in the northern half of the study area, requiring low temperatures, whether on average throughout the

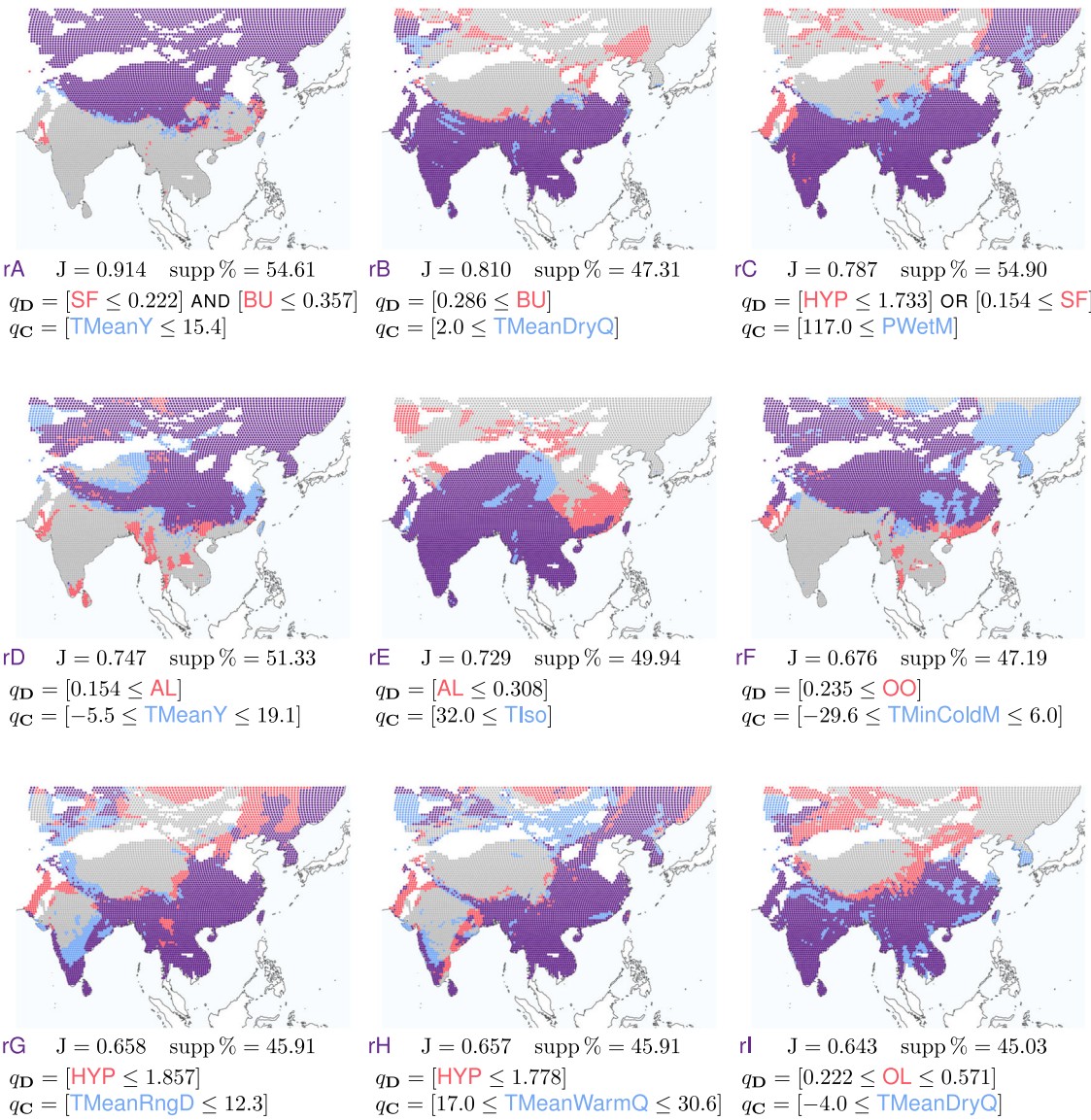

**Fig. 1 | Redescriptions rA–rI in the present-day dataset.** Localities that satisfy both queries, only the dental traits query, only the climate query, and neither query, are drawn in purple, red, blue, and gray, respectively. For each redescription, we list the query over dental traits variables ($q_D$), the query over bioclimatic variables ($q_C$), the accuracy (J) as well as the size of its support as a percentage of the total number of localities (supp%). See maps visualization in ref. 24.

year (rA: [TMeanY ≤ 15.4] and rD: [−5.5 ≤ TMeanY ≤ 19.1]) or at the lowest of the coldest month (rF: [−29.6 ≤ TMinColdM ≤ 6.0]). On the other hand, redescriptions rB and rI select regions in the southern half of the study area, requiring mild to warm temperatures. Specifically, the average temperature should not drop below moderate during the driest quarter (rB: [2.0 ≤ TMeanDryQ] and rI: [−4.0 ≤ TMeanDryQ]). Overall, the cold climate of the northern half is associated with a low prevalence of bunodonty among the mammalian communities (rA: [BU ≤ 0.357]) and structural fortification of cusps (rA: [SF ≤ 0.222]) as well as a high prevalence of acute lophs (rD: [0.154 ≤ AL]) and exclusively obtuse lophs (rF: [0.235 ≤ OO]) among the mammalian communities. Vice versa, the warm climate of the southern half is associated with a high prevalence of bunodonty (rB: [0.286 ≤ BU]) and moderate prevalence of obtuse lophs (rI: [0.222 ≤ OL ≤ 0.571]) among the mammalian communities.

Three redescriptions (rC, rG, and rH) outline a southeast–northwest zoogeographic pattern across the study area. Redescription rC selects regions in the east and south of the study area that receive high precipitation amounts during the wettest month ([117.0 ≤ PWetM]), which

correspond to the regions under the influence of the Asian summer monsoon. Redescriptions rG and rH select more specifically East and Southeast Asian Monsoon regions, i.e., the same regions as rC but excluding inland India, requiring respectively limited diurnal temperature variations ([TMeanRngD ≤ 12.3]) and a mild warm season ([17.0 ≤ TMeanWarmQ ≤ 30.6]). In terms of dental traits, redescriptions rG and rH require a low prevalence of hypsodonty (HYP) among the mammalian communities, while rC requires a low prevalence of hypsodonty or a non-negligible prevalence of structural fortification of cusps (SF) (or both).

The remaining redescription (rE) does not fall squarely in either group, as it covers the Indian subcontinent and Southeast Asia, but also extends northward over the Tibetan plateau, requiring high isothermality, on one hand, and a low prevalence of acute lophs, on the other hand.

The climate queries of redescriptions rA and rC involve a couple of dental traits each, using respectively a conjunction and a disjunction. To properly understand such redescriptions, it can be useful to consider separately the subqueries involving each variable. Such

further examination (see maps visualization in ref. 24) reveals that the conditions on SF apply almost exclusively to India. Indeed, prevalences of structural fortification of cusps above 15.4% or even 22.2% are found throughout the subcontinent and hardly anywhere else. Furthermore, lower prevalences of bunodonty (BU ≤ 0.357) and higher prevalences of hypsodonty (1.733 < HYP) are commonly encountered today in inland India, unlike in East China and Southeast Asia.

Next, we take some of these redescriptions from the present day and evaluate them on our dataset from the past, which combines data from the fossil record with paleoclimate model simulations, aiming to track the emergence of these modern zoogeographical regions during the Neogene.

After evaluating them on the past dataset, we can visualize the status of each redescription for fossil localities from a given time interval as a map, using the same color code. For reference, we show the status of the redescription in the present day in the background. We show the maps for redescriptions rB and rC in Fig. 2. Further maps are provided in ref. 24.

## Emergence of the modern north–south zoogeographic division

Redescriptions rA and rB best capture the north–south zoogeographic division of the present day. In fact, the two redescriptions are almost logical complements of one another and tell essentially the same story, despite some differences. Thus, we focus on rB which lends itself to easier interpretation, thanks in particular to its simpler dental traits query.

In Fig. 2a, we see that fossil localities at low and middle latitudes are drawn in purple, meaning that the conditions at these localities satisfy both the dental traits and climate queries of redescription rB. In other words, fossil localities from the Early Miocene (23.03–15.97 Ma) in that area match the conditions of modern tropical and subtropical areas as captured by rB. On the other hand, fossil localities at high latitudes are drawn in gray, meaning that cold winters and low fractions of bunodont species are encountered there. This indicates that a north–south division already existed then. Blue and purple dots represent localities that satisfy the climatic conditions of the redescription. Such localities appearing northward of the present-day support of rB (i.e., falling on a gray background to the north of the purple background) suggest that warm hospitable conditions in the Early Miocene extended further north than they do today.

The north–south pattern is also visible in the Middle Miocene (15.97–11.63 Ma) (Fig. 2b). We note the presence of several red dots and a single blue dot among the gray dots at high latitudes. This means that the fossil communities have a fraction of bunodont species above the threshold specified in the dental query, that the redescription associates with warmer conditions, while the climate model predicts relatively cold temperatures, below the threshold specified in the climate query, for most localities at high latitudes. From the faunal perspective, the bunodont mammals seem to have achieved maximum geographic extent during the Middle Miocene.

In the early Late Miocene (11.63–7.246 Ma) (Fig. 2c), the northern boundary of rB has retreated southward, with the fossil localities supporting rB (purple dots) falling exclusively within its support in the present-day data (purple background). Localities outside this area are mostly gray, with a few exceptions in blue, i.e., low prevalence of bunodonty but warmer temperatures. The northern boundary of rB in the early Late Miocene appears to have been similar to the present-day situation, suggesting that a north–south zoogeographic pattern comparable to the modern may have emerged before the Late Miocene.

The northern boundary of rB appears to be stable in the late Late Miocene (7.246–5.333 Ma) (Fig. 2d), and is also fairly clear in the Pliocene (5.333–2.58 Ma) (Fig. 2e), at least in terms of climate. The four blue dots in Southeast Asia correspond to high-altitude sites of the Hengduan Mountains in Yunnan (China) and Gwebin in Burma that appear to host mammalian communities with a low prevalence of

bunodonty, unlike the mammalian communities found in the same or neighboring sites during the Miocene.

## Emergence of the modern southeast–northwest zoogeographic pattern

Redescriptions rC, rG, and rH capture the southeast–northwest zoogeographic distributional pattern of the present day. In fact, species with structural fortification of cusps never exceed 12.5% of fossil communities in our dataset, and are entirely absent in most cases. Therefore, the condition on SF in the dental query of rC is not satisfied at any of the fossil localities, so that, when evaluated on the past dataset, redescription rC behaves as if the dental query consisted only of the condition on HYP. This means that wherever the dental query is satisfied (red or purple dots in Fig. 2f–j) the condition of low prevalence of hypsodonty holds true. Thus, the dental queries of the three redescriptions are very similar, all requiring low prevalences of hypsodonty, with somewhat different thresholds. We focus on redescription rC, which involves the lowest threshold, paired with a condition on precipitation (PWetM).

In Fig. 2f, g, we see that all but two dots are either red or purple, meaning that during the Early and Middle Miocene, nearly all fossil communities in our dataset exhibit low prevalences of hypsodonty. This is in contrast to the modeled climate conditions indicating that fossil localities receiving high amounts of precipitation during the wettest month (dots that are either blue or purple) are restricted mainly to the south of the study area

In the early Late Miocene (Fig. 2h), higher prevalences of hypsodont and mesodont taxa are found among fossil communities at middle latitudes, with overall more gray dots in northern China, while the boundary of the climate query appears to be fairly stable, spanning east–west across the study area.

In the late Late Miocene (Fig. 2i), localities satisfying the climate query are found further in northeastern China, with the boundary tilting counter-clockwise. The prevalence of hypsodonty also appears more contrasted across this boundary. In particular, a majority of localities have higher (resp. lower) prevalences of hypsodonty in the northwestern (resp. southeastern) part of East China, with more gray dots than red (resp. more purple dots than blue). This suggests that the distribution of precipitation and of hypsodonty changed from a primarily north–south pattern to a southeast–northwest pattern, similar to the monsoonal pattern found in the present-day data. Fossil data from the Pliocene is rather limited, but the pattern appears to persist (Fig. 2j).

## The dynamics of zoogeography in the context of climate and tectonics

To provide context for our results and shed more light onto the processes that might have impacted the Asian faunal distribution during the Neogene, we analyze existing quantitative proxy data of temperature, precipitation, and orographic height of the Tibetan Plateau, along with the dynamics of two main dental traits and two main bioclimatic variables within subregions of the study area, during the Neogene (Fig. 3).

Figure 3a shows the global temperature trend (based on ref. 25) along with the average and standard deviation of BU and TMeanY calculated over localities from northern and southern Asia (delimited as specified in Supplementary Table 3) for each of the five-time intervals. We see that the dynamics of bunodonty in Asia during the Neogene follow global temperature changes. The prevalence of bunodont species remains consistently higher in southern Asia than in northern Asia throughout the analysis period.

Figure 3b shows the modeled mean annual precipitation for East Asia (based on ref. 26) along with the average and standard deviation of HYP and PTotY calculated over localities from northwestern and southeastern China (delimited as specified in Supplementary Table 3) for each of the five-time intervals.

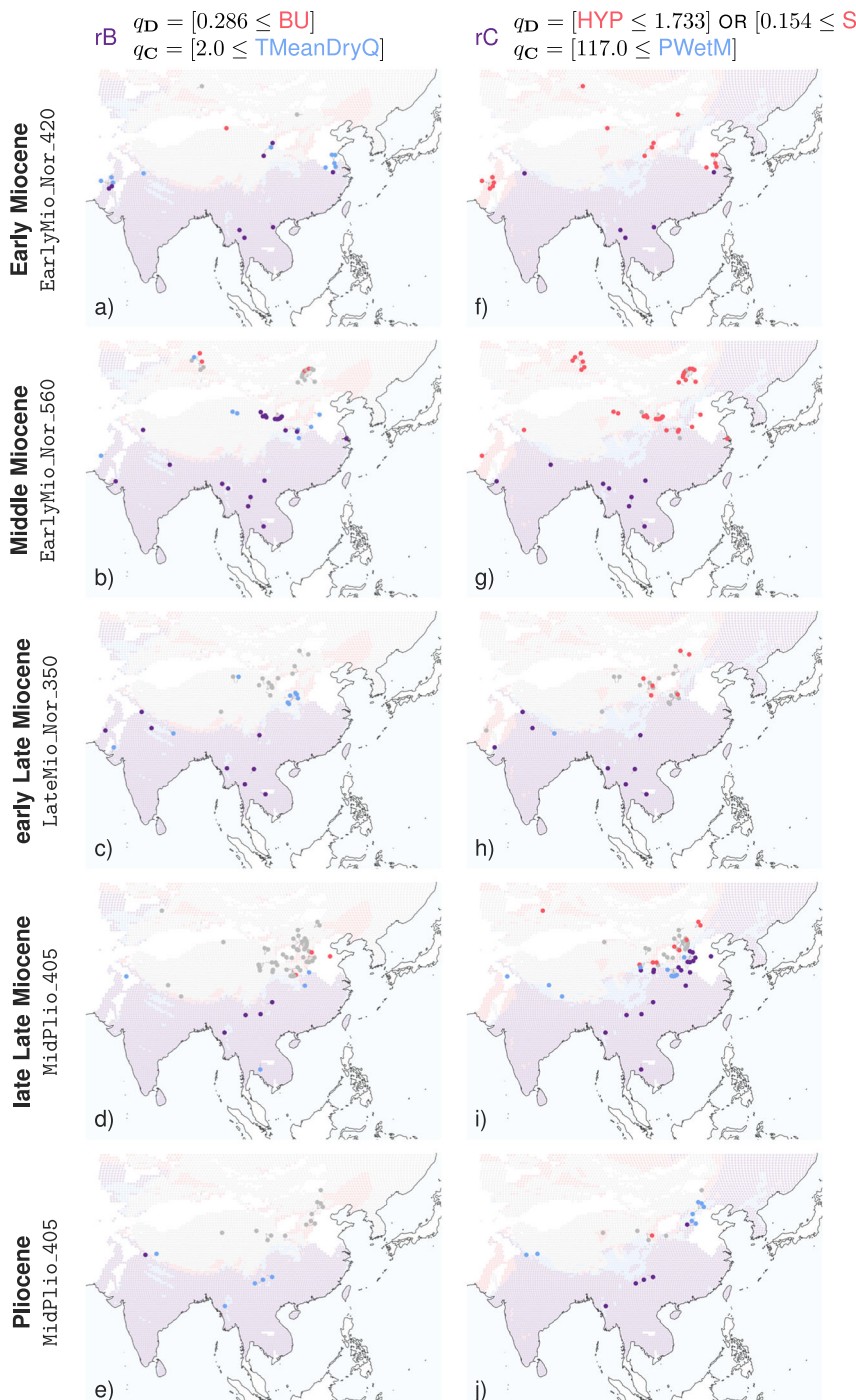

**Fig. 2 | Focus maps of redescriptions rB and rC (columns) evaluated on fossil localities from the different time intervals, considering the corresponding paleoclimate model simulation (rows). a** rB in the Early Miocene; **b** rB in the Middle Miocene; **c** rB in the early Late Miocene; **d** rB in the late Late Miocene; **e** rB in the Pliocene; **f** rC in the Early Miocene; **g** rC in the Middle Miocene; **h** rC in the early Late Miocene; **i** rC in the late Late Miocene; **j** rC in the Pliocene. Fossil localities that support both queries, only the dental traits query, only the climate query and neither queries, are drawn in purple, red, blue, and gray, respectively. Present-day localities are drawn in the background, for reference. See maps visualization in ref. 24.

We see that average hypsodonty and precipitation are clearly negatively correlated in southeastern China, in agreement with the expectation that higher hypsodonty is associated with more arid climates. On the other hand, in northwestern China the variables follow unsynchronized increasing trends, contradicting this expectation.

In addition, Fig. 3c shows elevation estimates obtained with either fossil-based or isotope-based methods for various sites of the Tibetan Plateau.

## Discussion

Among the climate constraints of the nine selected redescriptions (rA–rI), eight contain temperature variables and only one a precipitation variable, pointing to the strong relation between temperature and the present-day mammalian distribution in Asia. This is not unexpected, since temperatures present a strong latitudinal gradient on a large scale while the latitudinal diversity gradient is well known to be the strongest distributional pattern in the natural world[27]. When the

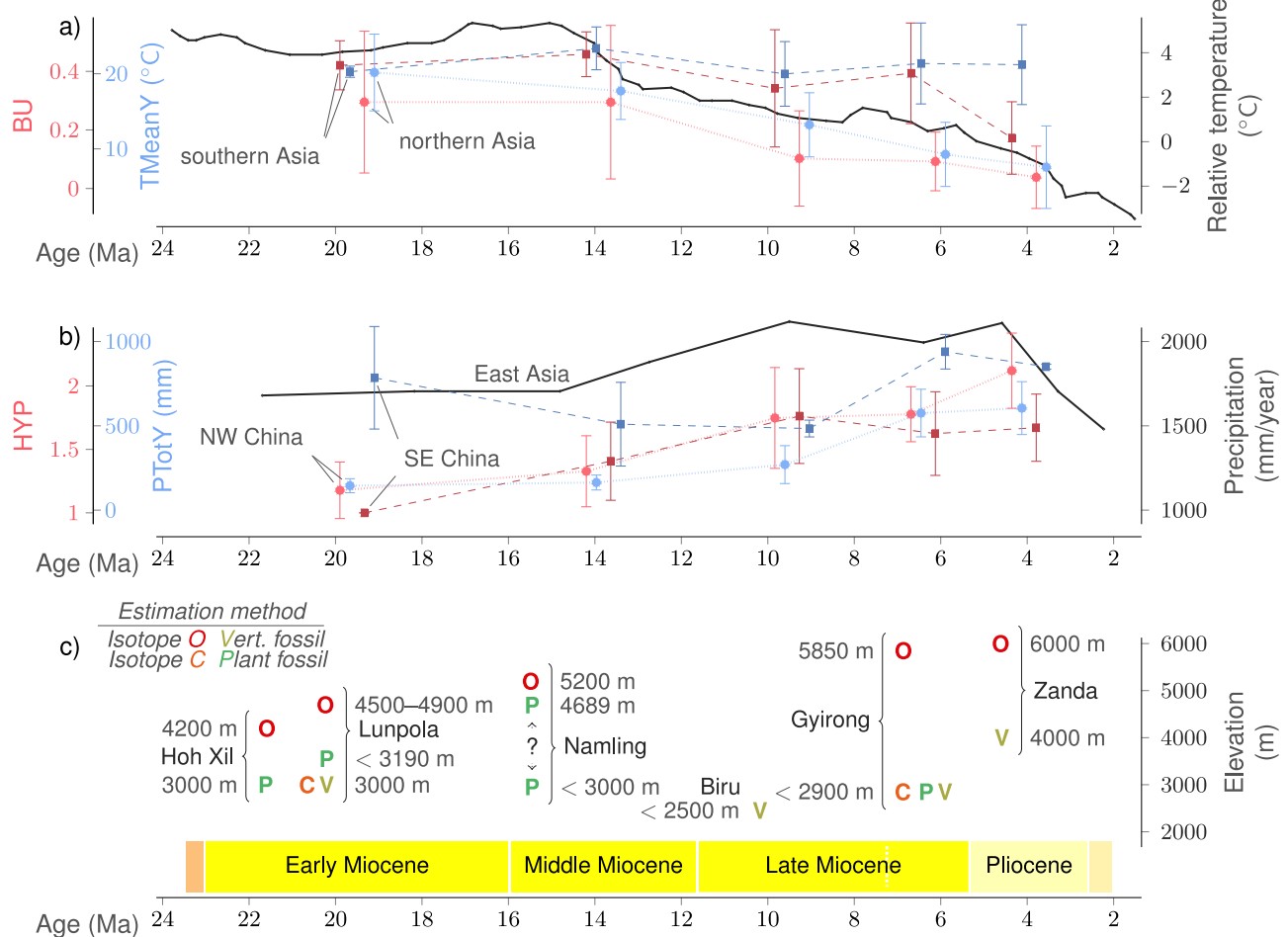

**Fig. 3 | Temperature, precipitation, elevation, bunodonty, and hypsodonty trends through the Neogene. a** Global temperature trend (based on ref. 25), along with bunodonty and mean annual temperature average values in northern and southern Asia. **b** Modeled mean annual precipitation for East Asia (based on ref. 26), along with hypsodonty and annual precipitation average values in northwestern (NW) and southeastern (SE) China. Average values and standard deviations (represented as error bars) are calculated over the localities in each group, which number (*n*) 18, 37, 23, 51, and 15 in northern Asia, 3, 8, 7, 5, and 4 in southern Asia, 6, 29, 14, 33, and 11 in NW China, 7, 8, 7, 20, and 6 in SE China, respectively for the five-time intervals. **c** Elevation estimates for the Tibetan Plateau (data resources in Supplementary Table 1).

modern latitudinal gradient was established is still an open question, however, especially within the terrestrial realm[28].

In this study, we are able to delineate a southern zoogeographic region through time via the association of mammalian bunodonty and a lower limit on the temperature of the driest quarter with redescription rB. The 0°C isotherm of the average temperature in January is one of the criteria separating the modern Palaearctic and Indomalayan realms[29]. The southern region satisfies rB, and such that [2.0 ≤ TMeanDryQ], hence corresponds well to the Indomalayan realm.

The prevalence of bunodonty (BU) was at its highest in both southern and northern Asia during the warm Early and Middle Miocene (Fig. 3a), and the northern boundary of rB lay further north than today (Fig. 2a, b). The maximum expansion of the warm and humid zone during the Middle Miocene coincides with the Middle Miocene Climate Optimum, around 17–15 Ma[25]. Following the significant cooling at the end of the Middle Miocene, the prevalence of bunodont species declined notably in both regions. Mammalian communities with high proportions of bunodonts disappeared permanently from high and middle latitudes and retreated to tropical and subtropical regions in the Late Miocene and Pliocene (Fig. 2c–e). This suggests that the modern Indomalayan realm was established most likely after the Middle Miocene Climate Transition, around 15–13 Ma[25,30], which is consistent with results from both marine fossil evidence[31] and phylogenetic modeling[15].

The increase of mean hypsodonty in the Middle Miocene relates to the immigration of hypsodont/mesodont rhinocerotids and bovids from West Asia and Europe[32]. This coincides with the appearance of grassland vegetation in Asia during the Middle Miocene[33]. The overall low hypsodonty in faunal communities in our data suggests grassland and dry conditions must have been very limited in East Asia during this interval[32,33].

The notable increase of mean hypsodonty in the early Late Miocene can be linked to the prevailing aridity at middle latitudes in East Asia[12] and the prominent expansion of grassland in the Asian inland[34,35]. Concurrent with the decline of bunodonty in Asia, the immigration and subsequent radiation of hypsodont *Hipparion* and bovids in Eurasia during the early Late Miocene[32,36] led to a substantial increase of hypsodonty (HYP) (Fig. 3b) and other traits associated with non-bunodont taxa (OL, OO, OT, and SF). High precipitation areas were limited to southern Asia during this interval (Fig. 2h). The occurrences of global mid-latitude aridity during the late Neogene have been interpreted as a result of the global climate cooling and are well documented in West Europe, America, East Africa, Australia, and China[12,30,37–42].

The high point of mean ordinated hypsodonty in southeastern China during the early Late Miocene is at odds with the simulated precipitation peak[26] (Fig. 3b). This apparent disagreement might be due to the difference in the considered regions. Fossil locality data

mainly come from localities lying at the middle latitudes of East China, whereas the simulations of Farnsworth et al.[26] are for the whole of East Asia.

During the Middle Miocene and early Late Miocene the prevalence of hypsodonty is comparable and undergoes a similar increase across mammalian communities of northwestern and southeastern China. From the late Late Miocene (ca. 7 Ma), a disparity appears between northwestern and southeastern China, which strengthens during the Pliocene. Relatively higher precipitation combined with lower mean ordinated hypsodonty start to dominate in southeastern China, in contrast to northwestern China where lower precipitation and higher average hypsodonty prevail. Such changes in regional precipitation regimes have been suggested to link to the onset or intensification of summer monsoon in East Asia and aridity in central Asia[32,43,44] as a result of the significant uplift or lateral extension of the Tibetan Plateau around this time[43,45].

In the Pliocene, the mammalian communities of the Hengduan Mountains and central Myanmar show a drop in the fraction of bunodont species (Fig. 2d, e), indicating that warm-favoring fauna disappeared from these areas. The low temperatures to which such mammalian communities point is consistent with the appearance of an altitudinal vegetation zonation[46–48] and the absence of previously prevailing middle-sized hominids in this area during the Pliocene[49]. Today's Hengduan Mountains are identified as a montane forest biodiversity hotspot[6], our results support the emergence of this hotspot during the Pliocene[50]. In the Pliocene, the fossil communities of the Tibetan Plateau are characterized by species endemic to the central Plateau[51], on one hand, and snow-adapted Zanda fauna in the southern Plateau[52,53], on the other hand, indicating that Alpine fauna had already established on the Tibetan Plateau. The contemporaneous presence of Alpine fauna on the Tibetan Plateau and a hotspot of montane forest on its southeast extension in the Hengduan mountains suggests that the modern faunal diversity in Asia has been fully developed since the Pliocene. Our result is consistent with the analysis of biodiversity based on phylogenetic modeling and bridges the gap between phylogenetic and fossil evidence[5].

Our study further highlights that modern Asian distribution has been shaped by global climate change and the tectonic evolution of the Tibetan Plateau. While the impact of global climate changes is rather clear, the impact of the Tibetan Plateau on the Asian biota and climate during the Neogene is difficult to trace because the orogenic history of the Tibetan Plateau is the subject of much controversy, involving both biotic and abiotic evidence[54–57]. Most of the evidence of a high Plateau relies on oxygen-stable isotope paleoaltimetry estimations, but this method comes with many uncertainties[58,59]. The differences between elevation estimates obtained with oxygen isotope and other proxies are considerable, with the former yielding much higher estimates (Fig. 3c). Looking through the lens of dental traits and climate redescriptions, the Tibetan Plateau started to differ from neighboring regions on its southeast extension (e.g., Yunnan), being distinctively colder, only from the early Late Miocene (Fig. 2b, c). This supports a relatively low Tibetan Plateau during the Early Neogene and is consistent with most of the biotic evidence[54].

Redescription mining captured patterns of association between bunodonty and warm conditions (rB), on one hand, and between low hypsodonty and high precipitation (rC), on the other hand, from the present-day dataset. While these associations hold relatively robustly in the late Neogene, they seem much weaker when evaluated on the data from the early Neogene. Many localities at middle latitudes have a low prevalence of bunodonty but mild temperatures during the dry quarter in the Early and Middle Miocene (blue dots in Fig. 2a, b), and vice versa for several localities at higher latitudes in the Middle Miocene (red dots in Fig. 2b). The disagreement between mean ordinated hypsodonty plotted over time and simulated precipitation seems even more prominent, with practically all of the localities from the northern

half of the study area having a low prevalence of hypsodonty but also having fairly dry simulated climate, in the Early and Middle Miocene (red dots in Fig. 2f, g). These disagreements might be due to a combination of reasons, including the following two.

First, the modern patterns of associations between dental traits and climatic conditions might not yet have applied in the early Neogene, not least because some of the dental traits characteristic of current large mammal communities were still in the early stages of evolution. Indeed, in the Early Miocene the fossil localities in question (blue dots in Fig. 2a) host communities with dominant non-bunodont perissodactyls (tapirs, chalicotheres, rhinos), small-sized deers and moschids. Considering the nearest living relatives, an evolutionary perspective that should also be taken cautiously, such fauna suggests warm conditions[60], despite consisting of a low proportion of bunodonts. Higher bunodonty is generally associated with low seasonality, but its relation to temperature in those times might not have been analogous.

From the Middle Miocene onwards, perissodactyls presumably favoring warm and humid conditions decreased abruptly[61], while bunodont primates, suids, and elephants occupied the forest habitats[32], forming high bunodont and low hypsodont mammalian communities. The same patterns of association between dental traits and climatic conditions as today are expected to apply to them. The warm and humid conditions in the high latitudes during the Miocene are not only supported by the presumably forest-adapted taxa *Pliopithecus*, *Kubanochoerus,* and *Gomphotherium*[62–66], but there is also ample evidence of warm and humid conditions related to the Middle Miocene Climatic Optimum at high latitudes and inland areas[34,67–69]. Hence, this first explanation is plausible for the Early Miocene, but not so later on.

Second, the simulated climate variables might not adequately reflect the prevailing conditions not least due to large uncertainties in the prevailing physical boundary conditions (e.g., atmospheric carbon dioxide, orbital forcings, geography, topography, and vegetation), especially in the early Neogene. Indeed, prominent paleoclimate models have been found to underestimate temperatures at high latitudes for the middle Miocene[70], which matches our observation. Furthermore, when looking at the predictions of paleoclimate models, it is generally understood to be more meaningful to consider relative values and trends of changes rather than absolute values, which are more difficult to predict reliably. A bias-correction procedure is applied to the predictions of paleoclimate models, increasing the robustness of the obtained values. Yet, because climate queries contain strict thresholds on absolute values, their evaluation in the past can be sensitive to systematic deviations that might persist in the predictions. This is not such a problem with dental traits, which for the most part are fractional values.

The combination of dental traits and climate is necessary and valuable as it enables us to automatically identify the relevant variables and thresholds and to extract meaningful queries from the present-day dataset thanks to redescription mining. Here, when using the redescriptions to reason about the past, we relied more heavily on the dental traits queries, even though such data is of course not free of uncertainties and potential biases. This approach allowed us to shed light on the timing of the emergence of modern zoogeographic regions in Asia. Further studies are needed to resolve the discord between dental traits and climate simulations, especially in the Early and Middle Miocene, which hopefully would shed more light not only on the biogeographic patterns during the early Neogene in Asia, but also provide further methodological insights for modeling mammalian communities and climate.

## Methods
### Study area and time intervals
Similarly to our previous study on the present-day ecoregions of China and Southern Asia[22], the area studied in this work is focused on China,

**Table 2 | Time intervals and paleoclimate model simulations**

| Time interval | | Nb. localities | Climate simulation | Model | Time | pCO$_2$ |
|---|---|---|---|---|---|---|
| Early Miocene | 23.03–15.97 Ma | 23 | EarlyMio_Nor_420 | NorESM | 20 Ma | 420 ppm |
| Middle Miocene | 15.97–11.63 Ma | 42 | EarlyMio_Nor_560 | NorESM | 20 Ma | 560 ppm |
| early Late Miocene | 11.63–7.246 Ma | 27 | LateMio_Nor_350 | NorESM | 10 Ma | 350 ppm |
| late Late Miocene | 7.246–5.333 Ma | 56 | MidPlio_405 | CCSM4 | 3 Ma | 405 ppm |
| Pliocene | 5.333–2.58 Ma | 17 | MidPlio_405 | CCSM4 | 3 Ma | 405 ppm |

the Indian subcontinent, and Southeast Asia, which have a distinct climate system, unlike any other region in the world.

We extend our study area from 40°N to 50°N to cover most of Northern China, a region that was excluded from our previous research but comprises rich Neogene fossils localities. We obtain the fossil data for countries in this area, namely China, India, Pakistan, Bangladesh, Sri Lanka, Burma, Laos, Buhdan, Cambodia, Thailand, and Vietnam. The fossil data span from the Early Miocene to the Pliocene (23–2.5 Ma), split into five-time intervals (Table 2), namely Early Miocene (23.03–15.97 Ma, 23 localities), Middle Miocene (15.97–11.63 Ma, 42 localities), early Late Miocene (11.63–7.246 Ma, 27 localities), late Late Miocene (7.246–5.333 Ma, 56 localities), and Pliocene (5.333–2.58 Ma, 17 localities).

The datasets of modern species occurrences and of bioclimatic variables (sites × species and sites × climate, respectively) come from ref. 18. We use square grid cells of 50 × 50 km as units of analysis.

### Species occurrence data
Present-day species occurrence data originally come from the list of the International Union for Conservation of Nature (IUCN, https://www.iucn.org/) processed by Oksanen et al.[18].

Fossil occurrence data were downloaded from the *New and Old Worlds Database of Fossil Mammals* (NOW, https://nowdatabase.org/)[71]. For the purpose of this study we updated the Asian fossil data in NOW following published literature.

We focus on the large herbivorous mammals which we select by taxonomic orders. That is, we focus on Perissodactyla, Artiodactyla, Primates, and Proboscidea. We only consider sites (present-day grid cells and fossil localities) that report at least three species of large mammalian herbivores.

### Dental traits data
The functional characteristics of the teeth of plant-eating mammals, such as crown height, scale isometrically with the size of the animal[72], allowing to directly describe animal communities in terms of the distribution of their functional dental characteristics. The most common macroscopic characteristic is hypsodonty, which describes how tall a tooth is in relation to its width or length. The more hypsodont, the more durable to wear the tooth is. The mean hypsodonty of a community has been widely used as a proxy for precipitation[8]. The proxy predicts precipitation primarily, although not exclusively, due to hypsodonty being common in grass-dominated ecosystems or otherwise open habitats but rare in (temperate) woody habitats, and open habitats in turn being typically drier than woody habitats. Pointed or rounded structures on the surface of a tooth are called cusps, while ridges of enamel connecting cusps are called lophs. Bunodonty refers to a dental morphology with separate cusps that are not fused into elongated, lophlike structures. This dental morphology is typical for omnivores and frugivores, including most suids and primates. Faunal communities with a high proportion of bunodont species are typically found in relatively warm and humid forested environments that lack seasonal differences. Another characteristic commonly used to estimate climatic conditions is the presence of longitudinal lophs. Acute lophs and obtuse lophs respectively designate sharp edges and blunt edges across the chewing direction. Globally, the greater the

prevalence of taxa having lophs in the community, the cooler the temperature is expected to be[19] in the harsh season. High average loph counts are typically associated with cold habitats and plant food which is harder to chew (especially during the harsh season). Structural fortifications are reinforcement of cusps making them more prominent and resistant to dental wear. Occlusal topography refers to the surface of a teeth being flat or non-flat.

Because the redescriptions from our previous work[21] could not capture the specificities of South–East Asia sufficiently well, here we consider an extended tailored set of dental traits. The core scoring scheme follows Žliobaitė et al.[17], but we score acute lophs (AL) for selenodonts as introduced by Oksanen et al.[18], and include two extra traits, namely bunodonty (BU) and exclusively obtuse lophs (OO) as described by Saarinen et al.[23]. Both traits can be uniquely determined from the other traits[17]. Specifically, BU is positive if none of AL, OL, OT, and SF is, that is, there are no lophs of any kind. OO, as the name suggests, is positive if OL is positive, but none of AL, OT, and SF are. All in all, we use seven dental traits variables to describe morphological characteristics of molar teeth: hypsodonty (HYP), bunodonty (BU), presence of acute lophs (AL), presence of obtuse or basin-like lophs (OL), structural fortification of cusps (SF), flatness of occlusal topography (OT), and exclusively obtuse lophs (goat-type) (OO). HYP is ordinal with three distinct values, whereas the other variables are binary (the characteristic is either present or absent).

We start from the scores for present-day dental traits from Galbrun et al.[22], which we update, including scoring AL for selenodonts and additional traits of BU and OO. We score dental traits for the fossil species specifically for this study. Dental traits are primarily scored at the species level from images in the literature and museum photos from our own archives. When no information for a particular species is available, we assign the dominant score of the genus, tribe, subfamily, or family that the species belongs to, as relevant. The scores used in this study are provided in ref. 24.

To prepare the datasets of dental traits for sites as needed for mining and evaluating redescriptions, we average individual dental traits of species occurring at each site. Hence, we obtain a dataset with seven dental traits variables (Table 1), such that each variable takes values in the unit interval and records the fraction of species in the mammalian community of the considered site in which the characteristic is present, with the exception of HYP, which is the average over the occurring species of the ordinal values.

### Climate data
On the climate side, we use the standard nineteen bioclimatic variables known as *Bioclim* (Table 1).

The modern observational data (*WorldClim2*, http://www.worldclim.com/version2) are originally based on Fick and Hijmans[73]. We prepare the present-day dataset following the methodology described in Galbrun et al.[22]. In particular, the data are mapped to the same 50 × 50 km grid to match the species occurrences and dental traits data.

To prepare the past climate dataset, we used paleoclimate models to simulate climate data, specifically monthly precipitation, and maximum and minimum temperature (averaged over multiple years), from which the bioclimatic variables are derived and interpolated for each fossil locality.

For each time interval, we select a paleoclimate model and associated parameters, as indicated in Table 2. In particular, the *Norwegian Earth System Model* (*NorESM*)[74] is used to obtain global climate model simulations at 20 Ma with pCO$_2$ of 420 ppm (denoted as `EarlyMio_Nor_420`) and 560 ppm (denoted as `EarlyMio_Nor_560`), and at 10 Ma with pCO$_2$ of 350 ppm (denoted as `LateMio_Nor_350`), for the Early Miocene, the Middle Miocene and the early Late Miocene, respectively. The same mid-Pliocene (3 Ma) simulation (`MidPlio_405`), obtained using the *Community Climate System version 4* (*CCSM4*)[75] with data provided by *ecoClimate*[76] (https://www.ecoclimate.org) is selected for both the late Late Miocene and the Pliocene.

To probe the impact of the choice of paleoclimate model and parameters on the results, we considered additional simulations, including using the global climate model *ECHAM5/MPIOM*[77] and the regional climate model *COSMO-CLM*[44,78]. In particular, *COSMO-CLM* was run with different elevations of the Asian mountain ranges, to perform sensitivity experiments. The elevation was varied from low (250 m altitude) to high (present-day height, above 5000 m altitude) either for all Asian mountain ranges, to represent their bulk uplift through the geological periods, or for selected mountain ranges (e.g., southern Tibetan Plateau, central Tibetan Plateau, Zagros, and Tianshan-Altai Mountains), to represent the outward growth of the Tibetan Plateau. In these experiments, both the Late Miocene and the present-day global climate data were used as lateral boundary conditions to better cover the uncertainties of global climate forcing in simulating the Miocene and Pliocene climate in Asia. In total, we considered twenty-nine paleoclimate model simulations (cf. ref. 24 for further details).

We downscale and calibrate the climate model simulations following a procedure similar to previous studies (e.g., refs. 73,76,79), aiming to limit and correct potential biases. First, we compute the anomalies between each simulation and the baseline climate simulation (present-day control run) for each variable at its original spatial resolution. Second, we bilinearly regrid the anomalies to the 10 × 10 min grid used by the baseline present climate data of *WorldClim2* (http://www.worldclim.com/version2) with tools provided by *Earth System Modeling Framework* (ESMF version 8.0.0, https://earthsystemmodeling.org, https://github.com/esmf-org/esmf/releases/tag/ESMF_8_0_0). Third, we apply the regridded anomalies to the baseline present-day climate data from *Worldclim2*.

Next, the nineteen bioclimatic variables are derived from the resulting monthly climate data, using the `biovars` function of the `dismo` R package (version 1.1–4, https://cran.r-project.org/web/packages/dismo/). Finally, the bioclimatic variables are interpolated to each fossil locality based on the nearest neighbors.

### Redescription mining

In this study, we are looking for associations between dental traits and climate conditions. In this context, a redescription provides two sets of constraints, called rules or queries, expressed as thresholds over dental traits variables and over bioclimatic variables, respectively. Given a dataset, each query is associated with the collection of sites where the constraints are satisfied. A pair of queries, respectively in terms of dental traits and of climate conditions, that are satisfied roughly at the same sites, can be interpreted as indicating the existence of a local pattern of association between the variables and is called a redescription. The more similar the two collections of sites, the stronger the pattern. This is the intuition behind the redescription mining task, which aims at identifying and statistically evaluating such patterns of associations between variables in a dataset. Different algorithms have been proposed[80] for this task. Here, as in our previous work[21,22], we perform the analysis with the `Siren` interface[81], using the `ReReMi` algorithm[82]. This greedy algorithm constructs the queries step by step. It generates conditions by selecting a variable and setting the

associated thresholds, using as candidates the values that appear in the data and aiming to maximize the redescription accuracy. For more information about these tools and about the connections between redescription mining and more classical methods such as regression and clustering, in the context of biogeography, we refer the reader to our previous work[22].

In the context of this study, each redescription consists of a set of constraints over dental traits variables, the dental traits query denoted as $q_D$, and a set of constraints over bioclimatic variables, the climate query denoted as $q_C$. An example redescription is given in (1)

$$q_D = [SF \leq 0.222] \text{ AND } [BU \leq 0.357] \qquad q_C = [TMeanY \leq 15.4], \qquad (1)$$

which reads as (note that redescriptions are bidirectional) "sites where the fraction of species with structural fortification of cusps (SF) is lower than 22.2% and the fraction of bunodont species (BU) is lower than 35.7% also often have a mean annual temperature (TMeanY) lower than 15.4 °C, and vice versa".

In the present-day dataset, 3497 grid cells satisfy both the dental traits constraints and the climate constraints. In other words, the redescription holds true at 3497 grid cells (depicted as purple dots in Fig. 1 rA). This represents 54.61% of the 6406 grid cells in the dataset, which we call the relative support of the redescription and denote as supp%. In this dataset, a further 160 grid cells satisfy the dental traits constraints but not the climate constraints (depicted as red dots), and 170 grid cells satisfy the climate constraints but not the dental traits constraints (depicted as blue dots).

The accuracy of a redescription is measured by the Jaccard coefficient, denoted as J, that is, the number of sites where both queries of the redescription hold, divided by the number of sites where either of the queries hold. In this case, it is equal to 3497/(3497 + 160 + 170) = 0.914, which is quite close to the maximum value of one. Note that for a redescription to be highly accurate, the two sets of constraints do not have to be satisfied in a large proportion of the sites, but there should be few sites where one is satisfied but not the other.

In summary, our example redescription captures a strong pattern that holds in the northern half of the studied area of the present-day dataset, linking low fractions of species with structural fortifications as well as of species without any lophs, on one hand, and relatively low annual temperatures, on the other hand.

### Analysis protocol

Our analysis proceeds in two main steps. First, we extract and select redescriptions from the present-day dataset, following Galbrun et al.[22]. Second, we evaluate the redescriptions on the past dataset, examining which queries hold where at each of the different time intervals.

Rather than reusing the redescriptions from Galbrun et al.[22], we rerun the mining process on the present-day dataset. We do this in order to update the dataset with respect to the study area and the considered dental traits. Running the mining process on the updated present-day dataset, we obtained 151 redescriptions. The run took about half an hour on a commodity laptop. We ranked the resulting redescriptions by decreasing accuracy and filtered out redundant ones. That is, we removed any redescription having more than 90% of its support in common with a more accurate one. This left us with 34 redescriptions. Then we selected the most accurate redescriptions that offer a good coverage of the study area as well as of the different dental traits variables. Except for dental trait OT, which is not represented among accurate redescriptions, all dental traits appear in at least one of the nine most accurate redescriptions. By clustering and visualizing these redescriptions, we checked that they provide a good coverage of the study area. Henceforth, we hence focus our analysis on these nine selected redescriptions, denoted as rA to rI, and the corresponding patterns in the fossil data.

We put our dataset for the past together by selecting a collection of fossil localities and collating the corresponding values for the dental traits variables and the bioclimatic variables, as explained earlier in this section. The selected redescriptions were then evaluated on this dataset, to determine the status of each query at every fossil locality. In particular, for a given fossil locality and redescription, we can determine whether the distribution of dental traits over the mammalian community at the locality satisfies the constraints of the dental query, on one hand, and whether the modeled bioclimatic variables at the locality satisfy the climate query, on the other hand. Hence, for a chosen time interval and redescription, we can plot the status of the redescription at each of the fossil localities of the time interval on a map, using the same color code as explained above (i.e., purple, red, blue, and gray representing localities where both the dental traits and climate queries, where only the dental traits query, where only the climate query, and where neither queries are satisfied, respectively).

### Reporting summary

Further information on research design is available in the Nature Portfolio Reporting Summary linked to this article.

## Data availability

All prepared input data and results analyzed during the current study are available in ref. 24, where they can also be visualized as maps. Present-day climate data come from *WorldClim2* (http://www.worldclim.com/version2)[73], present-day species occurrence come from the International Union for Conservation of Nature (IUCN, https://www.iucn.org/), processed by ref. 18, and fossil occurrence data come from the *New and Old Worlds* Database of Fossil Mammals (NOW, https://nowdatabase.org/)[82].

## Code availability

To allow full reproducibility of our analysis, the scripts that fully automate the processing pipeline as described in the manuscript, along with a copy of the source data that we used, are available in ref. 24. For preparing paleoclimate model simulations, we used the *Norwegian Earth System Model* (*NorESM*)[74], the *Community Climate System version 4* (*CCSM4*)[75] with data provided by *ecoClimate*[76] (https://www.ecoclimate.org), the *Earth System Modeling Framework* (ESMF version 8.0.0, https://earthsystemmodeling.org, https://github.com/esmf-org/esmf/releases/tag/ESMF_8_0_0), and the `dismo` R package (version 1.1-4, https://cran.r-project.org/web/packages/dismo/). The data analysis was performed with the `Siren` interface[81], using the `ReReMi` algorithm[82] (https://gitlab.inria.fr/egalbrun/siren, commit f154c53b).

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

## Acknowledgements

We thank Thomas Denk from the Swedish Museum of Natural History for their critical reading of this manuscript and insightful comments. This work was supported by grants from the Research Council of Finland (314803 I.Ž., 316799 A.K., 341620 L.L., 341622 A.K., 341623 I.Ž.).

## Author contributions

E.G., H.T., A.K., and I.Ž. conceived the study; L.L and A.K. selected and curated fossil localities. L.L. coded fossil herbivore dental traits, L.L. and I.Ž. coded modern herbivore dental traits. H.T., Zhongshi Z., and Zijian Z. provided the data from paleoclimate model simulations. E.G. carried out the algorithmic analysis of the data and visualized the results. L.L. and E.G. led the writing of the manuscript. I.Ž. and H.T. contributed critically to drafts of the manuscript. All authors finalized the text.

## Competing interests

The authors declare no competing interests.
