## [Peer Review File · Nature Communications]

The emergence of modern zoogeographic regions in Asia examined through climate–dental trait association patternsReviewers' Comments:

Reviewer #1:

Remarks to the Author:

Dear Dr. Galbrun, et al.,

After reviewing your manuscript entitled "The emergence of modern zoogeographic regions in Asia through the lens of climate–dental traits association patterns", I believe this manuscript is worthy of publication. The authors should be commended for work that is well grounded in the previous literature, but is also novel in its methodology. To my knowledge, outside of the authors' own previous work establishing the method as a viable approach, redescription mining has not been applied to this type of data before. An expansion of the types of analyses used in paleogeography is a valuable contribution to the field.

Though I am not as familiar with the literature pertaining to the paleogeography of the Asian continent, the manuscript makes a compelling case for why their results are an important step forward in understanding the evolution of modern ecosystems. Their conclusions are thoughtful and well-grounded, and the supplemental materials provide a reasonable blueprint for reproducibility.

The suggestions that I provide in an attached, annotated PDF of their manuscript are solely in regards to the accessibility of the manuscript to a more general audience. I suspect that readers unfamiliar with the dental traits analyzed here, or with redescription mining, will find large portions of the manuscript to be inaccessible.

However, these relatively minor issues do not take away from the value of the work as a whole, and I fully endorse the publication of this manuscript in Nature Communications.

Sincerely,
Jackson Spradley

Reviewer #2:

Remarks to the Author:

This paper presents original research results that are an extension of previous work by this group. It entails a refined look at climate-dental trait associations on a large (Asian continent) scale. The authors look at continent-wide trends, latitudinal and correlated with monsoon systems, to draw inferences on the timing of development of current stratification. It should be published, but needs significant reworking and thought. It is very dense – one could read it quickly and gain nothing beyond the abstract – or one can spend significant time to understand details.

An example is the supplementary info: authors have dumped everything in there – it is not helpful or pleasant to behold tons of code – or the statement (Sorry about that, but we can't show files that are this big right now.) How about some explanation in plain language? Okay, I looked at a lot of the tabs; criticisms can quibble with details. Layout of tables is weird (columns don't line up). Ages are not justified (probably wrong in some cases) and these or geographic coordinates are cited at ridiculous levels of precision. Advice: just admit that the group has done the best it can and depends on reader input to ferret out inaccuracies.

Comments on the main article: Abstract – soften the claims: this paper suggests that by mid-Miocene the N-S pattern was in place. Great, but the data do not allow connection in time with the mid-Miocene climatic transition; why specify 7 Ma for the monsoon – that is not seen in the data; citation about Pliocene is sloppy – you mean since the Mio-Plio boundary

Line 34 contrasted line 37, delete 'a couple of'

As a criticism throughout, but I thought of it in Table 1 is that you depend on presence-absence (faunal lists) but ignore abundances

Fig. 1: I don't see dots! (but I do on fig. 2)

Line 92, delete 'where', use 'for' for represent

Line 94: neither query is satisfied, respectively. Purple represents support for each redescription.

Please say something like that or else I am completely lost

Line 107 – delete Vice versa Line 129 – use "which" for 'that' Line 144 delete 'a' Line 151 use "that"

for 'as' Line 153, delete 'high' Line 168: cups Delete line 213

Line 231 rhinocerotids.

Is the last sentence (lines 232-3) contradictory?

Line 258 – you are talking about the Miocene, not Pliocene

Line 262, you do not show 'coexistence'

Line 315: even though such data

Line 241 – here and elsewhere you do not tell us the size of grid applied

Line 417 – do you mean 'and vice versa'? Line 421 160 grid cells satisfy Line 422 170 grid cells satisfy.

You proceed to ignore the 45.39% Line 441 delete 'top'

September, 2023

Revision of Nature Communications manuscript NCOMMS-23-22781-T

Dear Reviewers,

thank you for the insightful comments and suggestions on our study.

We carefully took into account all remarks to revise our manuscript. In particular, we believe that the presentation of our work is improved in this revision, increasing the accessibility of our study to readers. We hope that you will agree. In addition to the edited manuscript, which we reformatted following Nature Communications guidelines, we provide a point-by-point response to the reviewers' comments (below), as well as a version of the manuscript with changes highlighted.

Sincerely,

The Authors

Reviewer #1 (annotated PDF)

p1: “However, the timings of how these distributional patterns arose”

⚡ REPLY: The sentence was edited to “However, the timings of the emergence of these distributional patterns”.

p2: How are the thresholds of a “query” determined/chosen?

⚡ REPLY: As mentioned in the following paragraph, redescription mining algorithms build the queries by iteratively extending candidate queries. In particular the thresholds are selected automatically, aiming to obtain pairs of queries that maximize the accuracy. We have added further explanations about how this is done in the Methods section.

p6: It would be helpful to have some reminder what variables are being considered by rB and rC in this figure (or its associated caption)

⚡ REPLY: We now list the queries at the top of the figure.

p8: The colors used in this figure are too similar, making it hard to follow the trends that you're trying to demonstrate.

⚡ REPLY: We use the same color code throughout, where red is associated to dental traits and blue to climate variable. In order to make the plots for the different regions easier to tell apart, we now use round vs. square markers and dotted vs. dashed lines.

p8: Why does the PTotY at the SE Chinese localities seem to contradict the modeled MAP for East Asia?

⚡ REPLY: We discuss this contradiction in the paragraph below (cf. lines 242–245 on page 9 of the reviewed manuscript): “The high point of mean ordinated hypsodontology in southeastern China during the early Late Miocene is at odds with the simulated precipitation peak [25] (Fig. 3b). This apparent disagreement might be due to the difference in the considered regions. Fossil locality data

mainly comes from localities lying at the middle latitudes of East China, whereas the simulations of Farnsworth et al. [25] are for the whole East Asia”.

p10: “yielding much higher estimates”

Higher estimates of what? I assume you mean elevation, but it’s not immediately clear.

⚡ REPLY: Indeed we mean elevation estimates. To make it clear, we edited the sentence to “The differences between elevation estimates obtained”.

p10: “they seem much weaker when evaluated on the data from the early Neogene”

Considering the discussion of the elevation of the Tibetan Plateau above, would these “weaker associations” result in a lower estimate of the elevation? RE: elevation estimates; What if the disagreement between stable isotope analysis and biotic evidence is due to a poor correlation between the chosen biotic evidence and the climatic variables in question?

⚡ REPLY: We argue that this is not due to lower elevation estimates for Tibet, but due to the disagreement between the modeled climatic variables and the dental traits proxy. We discuss this immediately after pointing out the disagreement.

p12: “All in all, we use seven dental traits variables . . . HYP is ordinal”

I would suggest summarizing these variables as they relate to a broad characterization of the diet of the species (high hypsodonty associated with grazing/browsing, high bunodonty associated with frugivory/gramnivory, etc.). I think this would go a long way in making the manuscript more accessible to a more general audience.

⚡ REPLY: We have added a more general description of dental traits and how they relate to diet and climate in the subsection titled “Dental traits data” of the Methods section.

p13: I’m not suggesting that you necessarily bother with this discussion in your manuscript, but I would be interested to hear the authors’ thoughts on the value of redescription mining versus the machine learning approaches I’ve used previously?

(See: Spradley et al., 2019. Mammalian faunas, ecological indices, and machine-learning regression for the purpose of paleoenvironment reconstruction in the Miocene of South America)

⚡ REPLY: As a major difference, redescription mining is a descriptive approach capturing patterns of associations between subsets of variables that hold locally, whereas machine learning approaches such as regression produce a global predictive model, assuming that the same relation between the variables holds throughout the dataset. We now mention this in the main text.

For a more in-depth discussion, we point to our previous work where we present a comparative case-study involving correlation analysis, regression and clustering besides redescription mining: Galbrun, E., Tang, H., Kaakinen, A. & Žliobaitė,

I. Redescription mining for analyzing local limiting conditions: A case study on the biogeography of large mammals in china and southern asia. *Ecological Informatics* 63, 101314 (2021). <https://doi.org/10.1016/j.ecoinf.2021.101314>

p14: Does the computer determine the thresholds of the queries that are then used to construct a redescription? If so, is there any way to describe how it comes to these values?

⚡ REPLY: Indeed, the thresholds are determined by the redescription mining algorithm. We have added a sentence in the Methods section with some further explanations:

“This greedy algorithm constructs the queries step by step. It generates conditions by selecting a variable and setting the associated thresholds, using as candidates the values that appear in the data and aiming to maximize the redescription accuracy.”

Reviewer #2

This paper presents original research results that are an extension of previous work by this group. It entails a refined look at climate-dental trait associations on a large (Asian continent) scale. The authors look at continent-wide trends, latitudinal and correlated with monsoon systems, to draw inferences on the timing of development of current stratification. It should be published, but needs significant reworking and thought. It is very dense – one could read it quickly and gain nothing beyond the abstract – or one can spend significant time to understand details.

An example is the supplementary info: authors have dumped everything in there – it is not helpful or pleasant to behold tons of code – or the statement (Sorry about that, but we can’t show files that are this big right now.) How about some explanation in plain language? Okay, I looked at a lot of the tabs; criticisms can quibble with details. Layout of tables is weird (columns don’t line up). Ages are not justified (probably wrong in some cases) and these or geographic coordinates are cited at ridiculous levels of precision. Advice: just admit that the group has done the best it can and depends on reader input to ferret out inaccuracies.

⚡ REPLY: We believe there might somewhat of a misunderstanding about what the code and data in the supplementary information are and why we provide them.

To put it simply, we provide code and data to allow full reproducibility of our analysis. Regular readers are certainly not expected to go through these materials in order to understand the analysis. But in case someone wants to run their own analysis with modifications, the code allows it.

The code that we provide fully automates the data processing pipeline as described in the manuscript. There are several scripts, representing a significant amount of code, simply because the data preparation process is fully automated

and must handle data from multiple source databases that come in slightly different formats, but it all boils down to loading the data records, checking which ones fall within the area and time intervals of interest and involve relevant species, then aggregating and formatting them as expected for the analysis, as we explain in plain language in the Methods section.

Our analysis rests on publicly available data sources, such as the NOW database. The teams behind these databases keep updating them, making edits, correcting and inserting records. Therefore, we include snapshots of the databases taken at the time we performed the analysis, so that our results can be replicated exactly by running the provided scripts. The large data files are not intended to be browsed online, they are meant to be processed automatically.

We provide step by step instructions on how to run the scripts to replicate the full analysis from scratch, all the way to reproducing the figures.

Providing the scripts also allows to rerun the analysis on more recent versions of the source databases, or with different choices for the time intervals, areas, taxonomic orders, etc., should anybody wish to do so. Admittedly, this requires some familiarity with the Python programming language in which the script are written. But this is definitely not necessary in order to understand our analysis and the discussion in the manuscript. In that sense, the manuscript is very much self-contained. Even if they are not necessary, and if they are useful to a limited fraction of the readers, we believe it is important to make all these materials available.

We wish to make more prominent the link to the visualization webpage where all prepared input data and results analyzed during the current study can be visualized as maps (<https://zliobaite.github.io/redescription-asia-neogene/>), i.e. where data and results can be browsed in a human-reader friendly form. Indeed it is likely to be most useful to the readership, as the main supporting resource for a more in-depth understanding of our analysis. Please note that guidelines about the visualizations can be accessed by clicking on the *i* in the top left corner of the page.

We have clarified the data and code availability statements and the GitHub readme files accordingly.

Along with the other clarifications added to the manuscript, we think that this improves the accessibility of our work, making it more readily understandable to the audience.

Comments on the main article:

Abstract – soften the claims: this paper suggests that by mid-Miocene the N-S pattern was in place. Great, but the data do not allow connection in time with the mid-Miocene climatic transition; why specify 7 Ma for the monsoon – that is not seen in the data; citation about Pliocene is sloppy – you mean since the Mio-Plio boundary

↩ REPLY: Indeed 5 Ma is a rough date, we do not want to point to the Mio-Plio boundary but some time during the Pliocene.

We agree that the dates mentioned in the abstract were approximative, and decided that it is better to remove them and edited the text accordingly.

We argue that the N-S pattern can be linked to the mid-Miocene climatic transition, relying on Figure 3a.

Line 34 constricted

↩ REPLY: The typo has been fixed.

line 37, delete “a couple of”

↩ REPLY: Deleted.

As a criticism throughout, but I thought of it in Table 1 is that you depend on presence-absence (faunal lists) but ignore abundances

↩ REPLY: The data that we use indeed does not contain abundances. Abundances are generally not available for fossil data at the continental scale and even if they were, relative abundances of fossils relate to many other factors than ecology, including taphonomy and collection biases.

Note that the present-day data also come in the form of faunal lists, which is consistent, as the present-day data from which the redescrptions are extracted must be in the same format as the fossil data on which they are evaluated, for the approach to be meaningful.

Fig. 1: I don't see dots! (but I do on fig. 2)

↩ REPLY: Fig. 1 shows the redescrptions in the present-day data. Each present-day locality, i.e. 50×50 km grid cell, is in fact depicted as a colored dot, but because there are many dots and they are quite small, one might overlook them and think that the picture is pixelated but this is not the case.

In Fig. 2 the fossil localities are displayed, they are fewer of them and we can use larger dots.

All maps can be viewed in online on the provided interactive visualization webpage where the individual localities are clearly visible and details for each one can be obtained by hovering over the corresponding dot (see <https://zliobaite.github.io/redescription-asia-neogene/>).

Line 92: delete “where”, use “for” for represent

Line 94: neither query is satisfied, respectively. Purple represents support for each redescription. Please say something like that or else I am completely lost

↩ REPLY: The explanations about the color scheme used in the maps have been edited for clarity.

Line 107: delete Vice versa

↩ REPLY: We actually want to emphasize that the dental traits conditions in the southern half are basically the converse of those in the northern half.

Line 129: use “which” for “that”

↩ REPLY: Edited.

Line 144: delete “a”

↪ REPLY: Deleted.

Line 151: use “that” for “as”

↪ REPLY: Edited.

Line 153: delete “high”

↪ REPLY: Deleted.

Line 168: cups

↪ REPLY: We meant dental cusps, the typo was fixed throughout.

Delete line 213

↪ REPLY: Deleted.

Line 231: rhinocerotids.

↪ REPLY: Changed rhinoceros to rhinocerotids.

Is the last sentence (lines 232–233) contradictory?

↪ REPLY: The sentence indeed sounds contradictory, we actually want to say that the dry condition in East Asia (Not Asia) must be limited. So we rephrase the sentence as “The overall low hypsodonty in faunal communities in *our data* suggests grassland and dry conditions must have been very limited *in East Asia* during this interval [31, 32].”

Line 258: you are talking about the Miocene, not Pliocene

↪ REPLY: Line 258 we cited “the disappearance of the last surviving middle-sized hominids in these areas, *Lufengpithecus* and *Khoratpithecus* [48]”. Indeed the disappearance of the last surviving middle-sized hominids may happened during the late Miocene. So, we edited the sentence to “The low temperatures to which such mammalian communities point is consistent with the appearance of an altitudinal vegetation zonation [45-47] and the absence of previously prevailing middle-sized hominids in this area during the Pliocene [48].”

Line 262: you do not show “coexistence”

↪ REPLY: We edited the sentence to “The contemporaneous presence of Alpine fauna on the Tibetan Plateau and a hotspot of montane forest on its southeast extension in the Hengduan mountains suggests that the modern faunal diversity in Asia has been fully developed since the Pliocene.”

Line 315: even though such data

↪ REPLY: Edited.

Line 241: here and elsewhere you do not tell us the size of grid applied

↪ REPLY: We use square grid cells of 50×50 km as units of analysis, as mentioned in Section “Study area and time intervals”.

Line 417: do you mean “and vice versa”?

↪ REPLY: Hyphen removed.

Line 421: “160 grid cells satisfy”

Line 422: “170 grid cells satisfy”. You proceed to ignore the 45.39%

↪ REPLY: We edited the sentence. In this paragraph we give an example to illustrate the definition of the relative support of a redescription. The relative support of a redescription is defined as the fraction of localities that satisfy both queries, which in this case is 54.61%, the remaining 45.39% are not ignored, they are just not part of the support of the redescription.

Line 441: delete “top”

↪ REPLY: Deleted.

Reviewers' Comments:

Reviewer #1:

Remarks to the Author:

The authors have made a commendable effort in improving the accessibility of the manuscript (my primary concern, as the research itself was sound in the original manuscript), and I am of the opinion that this manuscript is acceptable for publication without further review.

Sincerely,
Jackson Spradley

Reviewer #2:

Remarks to the Author:

Thank you for addressing my concerns. Technical actions that you have taken are appreciated. As to the dots on Fig. 1, you might simply note that they are very small at the reproduced scale. My comments about timing stem from a lack of fossil data in time - cause and effect cannot be demonstrated when events are separated by hundreds of thousands of years. There are VERY few fossil points relative to today. In South China, how many sites characterize the entire Miocene? As to data, most readers have to accept your analysis - I won't be trying to reproduce your results.

November, 2023

Final revisions for Nature Communications manuscript NCOMMS-23-22781A

Dear Reviewers,

thank you once again for the insightful comments and suggestions on our study. As far as we can tell, there were no outstanding comments requiring further edits of the manuscript.

We reproduce the reviewers' comments below, and answer the question that was asked.

Sincerely,

The Authors

Reviewer #1

The authors have made a commendable effort in improving the accessibility of the manuscript (my primary concern, as the research itself was sound in the original manuscript), and I am of the opinion that this manuscript is acceptable for publication without further review.

Reviewer #2

Thank you for addressing my concerns. Technical actions that you have taken are appreciated. As to the dots on Fig. 1, you might simply note that they are very small at the reproduced scale. My comments about timing stem from a lack of fossil data in time - cause and effect cannot be demonstrated when events are separated by hundreds of thousands of years. There are VERY few fossil points relative to today. In South China, how many sites characterize the entire Miocene? As to data, most readers have to accept your analysis - I won't be trying to reproduce your results.

↩ REPLY: As indicated in Table 1, the five time intervals contain respectively 23, 42, 27, 56 and 17 localities, i.e. 148 sites altogether for the Miocene and 165 sites in total. Note that every site belongs to a single time interval.

This is indeed a very small number when compared to the 6406 localities of the present-day dataset. But arguably that is expectable given the much easier access to present-day specimen than to fossil specimen.